# Knowledge Distillation for Semantically Inconsistent Data

## Abstract

Knowledge Distillation (KD) is a widely used technique for model compression, transferring knowledge from a large teacher model to a smaller student model. While most existing KD approaches focus on aligning outputs (logits) or intermediate features between teacher and student models, they typically overlook a key observation: not all training samples contribute equally to the knowledge transfer process. To address this limitation, this paper introduces a novel method to identify the data which exhibits semantic inconsistency between its input space and feature space. By adaptively assigning higher weights to these semantically inconsistent data during student model learning, the proposed method can refine the teacher's knowledge to better align with the student's needs, thereby improve the general knowledge distillation process. To demonstrate the general effectiveness of the proposed method, we embed it into several popular KD frameworks and extensively evaluate it on a diversity of teacher and student architectures. The experimental results prove that the proposed method can significantly boost knowledge distillation tasks, and set new state-of-the-art results on the CIFAR-100, Tiny-ImageNet, and ImageNet datasets. Code is given in the supplementary material.

## 1 Introduction

In recent years, deep neural networks (DNNs) have achieved impressive success in a wide range of tasks such as image classification (Krizhevsky et al., 2012), object detection(Ren et al., 2015), and natural language processing(He et al., 2016; Dosovitskiy et al., 2021). However, high-performing models like ResNet-152 or Vision Transformer(Dosovitskiy et al., 2021) typically come with massive parameter counts and computational burdens, making them difficult to deploy on real-world devices with limited resources, such as mobile phones or embedded systems.

To alleviate this problem, Knowledge Distillation (KD)(Hinton et al., 2015) has emerged as a popular and effective technique for model compression. The goal of KD is to transfer knowledge from a large, powerful teacher model to a lightweight student model, allowing the student to retain competitive accuracy while significantly reducing the computational cost. Many KD methods have been proposed over the years, including distillation through soft labels (logits)(Hinton et al., 2015), intermediate feature matching(Romero et al., 2015), attention alignment(Zagoruyko & Komodakis, 2017), and relational knowledge transfer(Park et al., 2019).

Despite their effectiveness, most existing KD methods assume that all knowledge from teachers is equally useful, and all training samples contribute equally to the distillation process (Muller et al., 2019; Kim et al., 2021a). This assumption is often unrealistic. In practice, not all the knowledge provided by the teacher model is meaningful or beneficial to the student, and some training samples are more informative or challenging than others (Cao et al., 2019). Treating all knowledge and all samples uniformly may result in inefficient knowledge transferring, especially when the student model has limited capacity and cannot absorb everything.

To address this issue, we propose a general technique to improve the existing KD methods. In particular, instead of assuming that all samples are equally important, our proposed method estimates how 'essential' each sample is by measuring whether its semantic relation in the **input space** is consistent with that in the **feature space**. A low semantic consistency suggests the corresponding data is challenging for student model, thus student model deserves more knowledge guidance from the teacher model when learning these data.

The main contribution of this paper can be summarized as follows:

- In contrast to most existing KD frameworks which minimize the overall output discrepancies between the student and teacher models on all the data, we try to identify the semantically inconsistent data and encourage the KD framework to mainly focus on the discrepancies on these data. By doing this, the student can be freed from the burden of aligning all its output to that of a strong teacher and is guided appropriately to distill only those truly important knowledge.
- We propose a novel mechanism to measure the semantic inconsistency of a data and adaptively embed this mechanism into various KD frameworks with several lines of codes.
- Extensive experiments on CIFAR-100, ImageNet and Tiny-ImageNet consistently demonstrate that our method can be easily incorporated into various KD frameworks and provide state-of-the-art (SOTA) performance.

## 2 RELATED WORK

Knowledge distillation (KD) has become a widely adopted technique for model compression(Hinton et al., 2015; Gou et al., 2021). The existing KD methods can be roughly grouped into two types: logit-based KD and feature-based KD.

**Logit-Based Knowledge Distillation.** Logit-based KD techniques aim to align the soft output distributions of the teacher and student models, initially proposed by(Hinton et al., 2015). Recent works enhance this paradigm by focusing on *dynamic temperature scaling* and *decoupled logit supervision*, both of which have demonstrated improved knowledge transfer efficiency.

(1) *Dynamic temperature scaling:* Traditional KD uses a fixed temperature hyper-parameter, which may limit the flexibility of the distillation process. To address this, methods like CTKD(Li et al., 2023) and MLKD(Jin et al., 2023) propose adaptive or multi-level temperature coefficients that better reflect sample difficulty and semantic depth. WTTM(Zheng & Yang, 2024) further improves performance by applying transformed teacher matching under dynamic temperature constraints.

(2) *Decoupled supervision:* Another line of work seeks to improve distillation by decomposing the logit information into finer components. ATS(Li et al., 2022) separates the KL divergence into multiple guidance terms (correction, smoothing, discriminability), while ReKD(Xu et al., 2024) explicitly partitions classes into head and tail for targeted supervision. SDD(Wei et al., 2024) restructures global predictions into local-level representations, enabling better fine-grained alignment.

**Feature-Based Distillation.** To better utilize the internal representations of deep networks, ReviewKD(Guo et al., 2020) introduces a review module that collects and reorganizes multi-stage feature maps from the teacher model. By transferring low- and mid-level features—often ignored by logit-based methods—ReviewKD offers fine-grained supervision, which can be helpful in the early training stage. This aligns with wider findings that feature-based distillation methods such as Attention Transfer (AT)(Zagoruyko & Komodakis, 2017) and more recent local contrastive methods like LCKA(Zhou et al., 2024) can improve the representational capacity of the student by preserving structural and spatial information in intermediate layers.

Recently, contrastive learning has been applied to KD as a way to better align the relational structures of feature spaces. Contrastive Representation Distillation (CRD)(Tian et al., 2020) reframes KD as a representation alignment task, encouraging the student to learn the same inter-sample similarity structure as the teacher. Following this idea, methods like DIST(Huang et al., 2024) incorporate dynamic intra-sample transfer and have demonstrated superior adaptation to instance-level difficulty.

## 3 METHODOLOGY

### 3.1 PRELIMINARIES AND MOTIVATIONS

To illustrate the original procedure of KD methods(Hinton et al., 2015), we consider a $C$-way classification task. Formally, let $D_{tr} = \{(x_i, y_i)\}_{i=1}^N$ be a training set, where each instance $x_i \in \mathcal{X} \subseteq \mathbb{R}^D$ has a corresponding class label $y_i \in \mathcal{Y} = \{1, 2, ..., C\}$, then a $C$-way classification task aims to learn

a model $\mathcal{M} : \mathcal{X} \to \mathcal{Y}$ from $D_{tr}$, so that $\mathcal{M}$ can achieve optimal classification ability. Typically, $\mathcal{M}$ can be divided into a feature encoder $f(\cdot)$ and a linear classifier $g(\cdot)$.

After $\mathcal{M}$ maps an input image $x$ to a logit vector $z \in \mathbb{R}^d$, i.e., $z = \mathcal{M}(x) = g(f(x))$, the probability of $x$ belonging to the $i$-th class can be calculated with Softmax function as follows,

$$p_i = \frac{\exp(z_i/T)}{\sum_{c=1}^{C} \exp(z_c/T)} \tag{1}$$

where $z_i$ denotes the $i$-th logit and $T$ is the temperature scaling hyper-parameter.

The original KD method aims to transfer the learned knowledge from a powerful teacher model $\mathcal{M}^t$ to a lightweight student model $\mathcal{M}^s$. While the teacher model has been well-trained from the training set $D_{tr}$, the student model $\mathcal{M}^s$ is trained from scratch by minimizing the Cross Entropy loss $\mathcal{L}_{CE}$ and Kullback-Leibler (KL) divergence loss $\mathcal{L}_{KD}$ as follows,

$$\mathcal{L}_{CE}(x_i, y_i) = -y_i \cdot \log(p_i) \tag{2}$$

$$\mathcal{L}_{KD} = KL(p^t || p^s) = \sum_{i=1}^{C} p_i^t \log(\frac{p_i^t}{p_i^s}) \tag{3}$$

where $p_i^t$ denotes the probability of $x$ belonging to the $i$-th class predicted by the teacher model $\mathcal{M}^t$. The overall loss $\mathcal{L}$ for the student model in the original KD task can be formulated as follows,

$$\mathcal{L} = \mathcal{L}_{CE} + w\mathcal{L}_{KD} \tag{4}$$

where $w$ is a hyper-parameter for balancing the two losses.

The original KD method significantly boosts student performance by minimizing the logit discrepancies between teacher and student, and such a strategy has inspired a lot of following works. Despite their effectiveness, the success of most KD frameworks has been built on the assumption that all training data are equally important for the knowledge transferring. Intuitively, this assumption can be easily violated in a real-world KD task. To this end, this paper attempts to identify the semantically inconsistent data, which we believe require the most urgent knowledge guidance from the teacher model.

### 3.2 KNOWLEDGE DISTILLATION ON SEMANTICALLY INCONSISTENT DATA

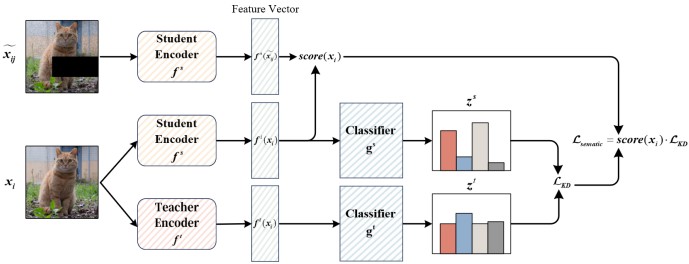

Figure 1: The pipeline of our proposed method. $x_i$ is the $i$-th data in a mini-batch, $\widetilde{x_{ij}}$ is obtained by mixing $x_i$ and $x_j$. The proposed score function $score(x_i)$ calculates the semantic inconsistency between feature $f^s(x_i)$ and $f^s(\widetilde{x_{ij}})$, and then assigns higher weights to the semantic inconsistent data, such that the improved KD loss $\mathcal{L}_{semantic}$ can provide more knowledge guidance on these data.

In this section, we will introduce our method on the basis of the original knowledge distillation, but as we will explain later, our method can be easily incorporated into various KD frameworks. Figure 1 shows the pipeline of the method. The core idea is to identify the semantically inconsistent data and assign higher weights to them, in such a manner, the KD procedure can focus on these data.

To quantify the semantic inconsistency of a specific data, we first apply strong data augmentation (i.e, Mixup (Zhang et al., 2018a)) on the data and then derive an inconsistency score based on the

similarity between the original data and its augmented counterpart. To be more specific, given a mini-batch of samples $\{(x_i, y_i)\}_{i=1}^{B}$, where $B$ is the batch size. Mixup generates synthetic sample $(\widetilde{x_{ij}}, \widetilde{y_{ij}})$ as follows,

$$\begin{cases} \widetilde{x_{ij}} = \lambda x_i + (1 - \lambda)x_j \\ \widetilde{y_{ij}} = \lambda y_i + (1 - \lambda)y_j \end{cases} \tag{5}$$

where $(x_i, y_i)$ and $(x_j, y_j)$ are the $i$-th and $j$-th samples drawn randomly from the mini-batch. $\lambda \in [0, 1]$ is a random value sampled from the Beta distribution $\mathbb{B}(\alpha, \alpha)$, with $\alpha \in (0, +\infty)$ as the hyper-parameter.

While Eq.(5) has frequently been used as a data augmentation method, an unexplored merit of Eq.(5) is that it actually constructs a semantic relationship between $x_i$ and $\widetilde{x_{ij}}$ in the **input space**, i.e., $\lambda$ *percent of $\widetilde{x_{ij}}$ is composed of $x_i$, and the remaining (1-$\lambda$) percent of $\widetilde{x_{ij}}$ is composed of $x_j$.* The model should also maintain a consistent semantic relationship in the **feature space**, otherwise, it indicates that the student model learns a poor representation of the data and the model deserves more knowledge guidance.

---

**Algorithm 1 Knowledge Distillation for Semantically Inconsistent Data**

---

**Require:** $D_{tr}$: training set; $\mathcal{M}^t, \mathcal{M}^s$: teacher and student models; $\alpha$: hyper-parameter of Mixup; $\beta$: base weight in Eq.(6);

---

**Train:**

1: **for** each training epoch **do**
2:     **for** each mini-batch $\{(x_i, y_i)\}_{i=1}^{B}$ in $D_{tr}$ **do**
3:         Generate data $(\widetilde{x_{ij}}, \widetilde{y_{ij}})$ with Mixup in Eq.(5),

$$(\widetilde{x_{ij}}, \widetilde{y_{ij}}) \leftarrow \text{Mixup}(x_i, y_i, \alpha)$$

4:         Feed $\{(x_i, y_i)\}_{i=1}^{B}$ and $\{(\widetilde{x_{ij}}, \widetilde{y_{ij}})\}_{i=1}^{B}$ to both teacher and student models, get the logits $z$ and feature $f$ as follows,

$$\widetilde{z_i^s}, \widetilde{f_i^s} = \mathcal{M}^s(\widetilde{x_{ij}})$$

$$z_i^s, f_i^s = \mathcal{M}^s(x_i)$$
$$z_i^t, f_i^t = \mathcal{M}^t(x_i)$$

5:         Calculate the $score(x_i)$ in Eq.(6) with $\widetilde{f_i^s}$ and $f_i^s$

$$score(x_i) = \beta - \frac{\exp(sim(f_i^s, \widetilde{f_i^s}))}{\sum_{j=1}^{B} \exp(sim(f_j^s, \widetilde{f_j^s}))}$$

6:         Calculate the standard KD loss $\mathcal{L}$ in Eq.(4)
7:         Perform backpropagation for the student model with the improved KD loss Eq.(7)
8:     **end for**
9: **end for**

---

Based on the insight described above, we propose a semantic inconsistency *score* as follows,

$$\begin{cases} score(x_i) = \beta - \frac{\exp(sim(f_i^s, \widetilde{f_i^s}))}{\sum_{j=1}^{B} \exp(sim(f_j^s, \widetilde{f_j^s}))} \\ f_i^s = \lambda f^s(x_i) + (1 - \lambda) \cdot f^s(x_j) \\ \widetilde{f_i^s} = f^s(\widetilde{x_{ij}}) \end{cases} \tag{6}$$

where $f^s$ is the feature encoder of the student model, $\widetilde{f_i^s} = f^s(\widetilde{x_{ij}})$ is feature representation of $\widetilde{x_{ij}}$ (i.e., the data generated by Eq.(5)), $f_i^s = \lambda f^s(x_i) + (1 - \lambda) \cdot f^s(x_j)$ is the linear combination of features $f^s(x_i)$ and $f^s(x_j)$, and $sim(f_i^s, \widetilde{f_i^s})$ calculates the cosine similarity between $f_i^s$ and $\widetilde{f_i^s}$ in the feature space. In addition, $\beta$ is the hyper-parameter that gives the base weight for each sample.

The second term (i.e., $f_i^s$) and third term (i.e., $\widetilde{f_i^s}$) in Eq.(6) are expected to be close to each other, so that the semantic relationship constructed by Eq.(5) can be maintained in the feature space of student models. Otherwise, a high score in Eq.(6) indicates that the semantic relationship in the **input space** has been violated in the **feature space** of the student model.

After the semantic inconsistency score in Eq.(6) has been calculated, the vanilla KD loss in Eq.(4) can be rewritten as follows,

$$\mathcal{L}_{Semantic} = \frac{1}{B} \sum_{i=1}^{B} score(x_i) \cdot \mathcal{L}(x_i) \tag{7}$$

Intuitively, the loss function in Eq.(7) is composed of two parts: (i) the semantic inconsistency score function $score(x_i)$, and (ii) the standard KD loss $\mathcal{L}(x_i)$. The score function assigns an adaptive weight to each sample $x_i$, so that the semantically inconsistent data receives more knowledge guidance during the KD procedure. Meanwhile, as the semantic inconsistency score is an independent term to the standard KD loss, the proposed method can be easily incorporated into various KD frameworks by replacing the standard KD loss $\mathcal{L}(x_i)$ with a more advanced KD loss.

**Computational complexity.** The exact steps of our proposed method are given in Algorithm 1. Three extra steps have been introduced by our method into a standard KD pipeline, and they have been highlighted in blue color. The extra steps can be implemented with several lines of codes, and they only introduce an extra forward passes of Mixup samples through the student model, which slightly increases training cost. Based on our experiments, the extra steps can increase the computational complexity of a standard KD pipeline by 20%-30%.

## 4 EXPERIMENTS

### 4.1 EXPERIMENTAL SETUP

**Datasets.** Three classic datasets have been taken as the benchmarks, including:

- **CIFAR-100**(Krizhevsky, 2009), which contains 100 classes, with 500 training and 100 testing images per class. The image size of CIFAR-100 is $32 \times 32$.
- **ImageNet** (Deng et al., 2009), which is the most widely used benchmark for large-scale visual recognition. Its training set contains roughly 1.3 million images of 1000 classes, and its validation set contains 50K images.
- **Tiny-ImageNet**(Le & Yang, 2015), which is a subset of ImageNet. It contains 200 classes and its resolution is $64 \times 64$. The training set of Tiny-ImageNet contains 100K images, while the testing set contains 10K images.

**Settings, Competitors and Performance Metrics.** We strictly follow the common practices in this field and focus on knowledge distillation in two different settings. (i) *Homogeneous architecture*, where the teacher and student share the same type of structure (e.g., ResNet110 and ResNet32) but differ in network capacity, and (ii) *Heterogeneous architecture*, where the teacher and student models adopt different architectures (e.g., ResNet and MobileNet). We consider a number of network structures, including ResNet(He et al., 2016), VGG(Simonyan & Zisserman, 2015), ShuffleNet-V1(Zhang et al., 2018b), ShuffleNet-V2(Ma et al., 2018), MobileNet-V2(Sandler et al., 2018) and WRN(Zagoruyko & Komodakis, 2016).

To prove the superiority of our method, we compare it with two groups of existing KD methods: (i) 11 feature-based methods, including RKD(Park et al., 2019), FitNets(Romero et al., 2015), AT(Zagoruyko & Komodakis, 2017), OFD(Heo et al., 2019), CRD(Tian et al., 2020), SRRL(Wang et al., 2021), ICKD(Wu et al., 2021), PEFD(Guo et al., 2023), CAT-KD(Chen et al., 2023), TaT(Yang et al., 2022), and ReviewKD(Guo et al., 2020), and (ii) 7 logit-based methods, including KD(Hinton et al., 2015), DKD(Zhou et al., 2022), CTKD(Cho & Hariharan, 2019), NormKD(Wang et al., 2023), Logit-Standardization-KD (LSKD)(Sun et al., 2024), MLLD(Zhao et al., 2023a) and CRLD(Zhang et al., 2024).

We also incorporate our proposed method into three representative methods (i.e., ReviewKD(Guo et al., 2020), CRLD(Zhang et al., 2024), and MLLD(Zhao et al., 2023a)). These three methods are

chosen not only because they are representative in both feature-based and logit-based KD paradigms, but also because they provide SOTA performance among existing baselines.

Top-1 and Top-5 accuracy are used as performance metrics. To ensure that the conclusions drawn from experimental results are reliable, all experiments are repeated three times, and we report the average performance.

**Implementation Details and Hyper-parameter Settings.** Following the common practices in this field, all student models are trained with SGD(Loshchilov & Hutter, 2019) with a momentum of 0.9 and a weight decay of $5 \times 10^{-4}$. For CIFAR-100, we set the batch size as 64, and the learning rate is set as 0.05. For ImageNet, the batch size is set as 256 and the initial learning rate is 0.05.

There are three hyper-parameters needed to be set before running the experiments, including: (i) $\alpha$ in Eq.(5), which generates new data with Mixup. We follow the default settings from the original paper, and set $\alpha = 0.2$ for CIFAR-100, and set $\alpha = 0.8$ for both ImageNet and Tiny-ImageNet. (ii) $\beta$, which is the base weight in Eq.(6). We set $\beta = 2$ in all our experiments. And (iii) temperature $T$ in the knowledge distillation. When combined with a specific KD method, we set the temperature $T$ to the default setting in their original papers. The influence of hyper-parameters $\alpha$ and $\beta$ will be discussed in the following section.

## 4.2 EXPERIMENTS RESULTS

**CIFAR-100.** We conduct extensive experiments on the CIFAR-100 dataset with homogeneous and heterogeneous teacher-student configurations. For fair comparison, we adopt the same experimental settings as prior works, and integrate our proposed method into three representative KD methods: ReviewKD, CRLD, and MLLD, resulting in three variants: **ReviewKD_Ours**, **CRLD_Ours**, and **MLLD_Ours**.

The results for homogeneous and heterogeneous setups are reported in Tables 1 and 2, respectively. In these tables, the best results for each setting are marked in bold font. As our method has been embedded into three different KD methods, the improvements brought by our method compared with the corresponding baselines are also reported. As shown in Table 1, when the teacher and the student share the same architecture, the combination of CRLD(Zhang et al., 2024) and our method (i.e., CRLD_Ours) provides the best performance. In particular, CRLD_Ours outperforms the best logit-based KD (i.e., CRLD) and feature-based KD (i.e., TaT(Yang et al., 2022)) by 0.7% and 1.79%, respectively. In addition, all embeddings of our method into the three existing KD methods bring improvements to the existing KD frameworks.

As shown in Table 2, when teacher and student adopt different network structures, our method also provides SOTA performance. Specifically, compared with the best logit-based KD (i.e., MLLD(Zhao et al., 2023b)) and feature-based KD (i.e., ReviewKD(Guo et al., 2020)), CRLD_Ours provides a performance gain of 0.94% and 1.42%. Meanwhile, the embeddings of our method into ReviewKD, CRLD and MLLD also bring performance gains.

The results report above not only demonstrate the general effectiveness of our method, it proves that our method can be easily embedded into various KD frameworks and bring essential improvements.

**ImageNet.** To further evaluate the generalizability of our method on large-scale datasets, we conducted experiments on ImageNet. As before, homogeneous and heterogeneous setups have been used. Specifically, ResNet34 and ResNet18 are taken as the teacher and student models in the homogeneous setting, while ResNet50 and MobileNet-V1 are taken as the teacher and student models for the heterogeneous setup. We compare our approach with several established baselines from both the logit-based and feature-based categories.

As shown in Table 3, in the homogeneous setting, our method MLLD_Ours provides the highest Top-1 accuracy, which outperforms the best baseline (i.e., LSKD) by 0.48%. In the heterogeneous setup, our method CRLD_Ours is the best method, which outperforms the best baseline (i.e., LSKD) by 0.9%. In addition, ReviewKD_Ours, which is obtained by incorporating our method in the baseline, have also brought performance gains, compared with the baseline KD frameworks (i.e., ReviewKD).

**Tiny-ImageNet.** When experimenting with Tiny-ImageNet, we adopt ResNet32x4 as the teacher model and ResNet8x4 as the student model. We compare our three improved variants, including

| Distill Type | Method | ResNet56 72.34 ResNet20 69.06 | ResNet110 74.31 ResNet32 71.14 | ResNet32×4 79.42 ResNet8×4 72.50 | WRN-40-2 75.61 WRN-16-2 73.26 | WRN-40-2 75.61 WRN-40-1 71.98 | VGG13 74.64 VGG8 70.36 | Avg. |
|---|---|---|---|---|---|---|---|---|
| Feature KD | RKD | 69.61 | 71.82 | 71.90 | 73.35 | 72.22 | 71.48 | 71.73 |
| | FitNets | 69.21 | 71.06 | 73.50 | 73.58 | 72.24 | 71.46 | 71.67 |
| | AT | 70.55 | 72.31 | 73.44 | 75.22 | 73.13 | 72.77 | 72.90 |
| | OFD | 70.98 | 73.23 | 75.63 | 75.24 | 73.55 | 74.33 | 73.83 |
| | CRD | 71.16 | 73.48 | 75.51 | 76.06 | 74.14 | 74.14 | 74.08 |
| | SRRL | 71.13 | 73.48 | 75.33 | 76.02 | 73.94 | 73.86 | 73.63 |
| | ICKD | 71.76 | 73.89 | 75.25 | 75.82 | 73.44 | 74.05 | 73.71 |
| | PEFD | 70.07 | 73.26 | 76.08 | 75.73 | 73.42 | 74.12 | 73.45 |
| | CAT-KD | 71.05 | 73.62 | 76.91 | 76.52 | 74.05 | 74.44 | 73.43 |
| | TaT | 71.59 | 74.05 | 75.89 | 76.42 | 74.35 | 74.58 | 74.15 |
| | ReviewKD | 71.89 | 73.89 | 75.63 | 76.16 | 74.12 | 74.67 | 74.06 |
| | ReviewKD_Ours | 71.93 | 74.15 | 75.74 | 77.02 | 75.51 | 74.90 | 74.87 |
| | Δ | +0.04 | +0.26 | +0.11 | +0.86 | +1.39 | +0.23 | +0.48 |
| Logit KD | NormKD | 71.40 | 73.91 | 76.40 | 74.84 | 74.49 | 74.48 | 74.25 |
| | DKD | 71.97 | 74.11 | 76.24 | 74.85 | 74.49 | 74.51 | 74.36 |
| | CTKD | 71.19 | 73.52 | 73.79 | 75.45 | 73.93 | 73.52 | 73.57 |
| | LSKD | 71.43 | 74.17 | 76.62 | 76.11 | 74.37 | 74.36 | 74.51 |
| | Logit-Standard-KD | 71.43 | 74.23 | 76.62 | 76.11 | 74.37 | 74.36 | 74.46 |
| | MLLD | 72.19 | 74.11 | 77.08 | 76.63 | 75.35 | 75.18 | 75.09 |
| | MLLD_Ours | **72.30** | 74.51 | 77.25 | 77.22 | 75.42 | 75.40 | 75.34 |
| | Δ | +0.11 | +0.4 | +0.17 | +0.59 | +0.08 | +0.32 | +0.25 |
| | CRLD | 72.10 | 74.42 | 77.60 | 76.45 | 75.58 | 75.27 | 75.24 |
| | CRLD_Ours | 72.20 | **75.06** | **79.12** | **77.25** | **75.91** | **76.18** | **75.94** |
| | Δ | +0.1 | +0.64 | +1.52 | +0.77 | +0.33 | +0.91 | +0.7 |

Table 1: Top-1 accuracy (%) on CIFAR-100 with homogeneous setting. The best results are marked with bold-font and Δ indicates the performance improvements brought by our method.

| Distill Type | Method | ResNet32×4 79.42 ShufV2 71.82 | ResNet32×4 79.42 WRN16-2 73.26 | WRN40-2 75.61 ResNet8×4 72.50 | WRN40-2 75.61 MobV2 64.60 | VGG13 74.64 MobV2 64.60 | ResNet50 79.34 MobV2 64.60 | Avg. |
|---|---|---|---|---|---|---|---|---|
| Feature KD | AT | 72.73 | 73.91 | 74.11 | 60.78 | 59.40 | 58.58 | 66.92 |
| | RKD | 73.21 | 74.86 | 75.26 | 69.27 | 64.52 | 64.43 | 70.26 |
| | FitNets | 73.54 | 74.70 | 77.69 | 68.64 | 64.16 | 63.16 | 70.65 |
| | CRD | 75.65 | 75.65 | 75.24 | 70.28 | 69.63 | 69.11 | 72.26 |
| | OFD | 76.82 | 76.17 | 74.36 | 69.92 | 69.48 | 69.04 | 72.30 |
| | CAT-KD | 78.41 | 76.97 | 75.38 | 70.24 | 69.13 | 71.36 | 73.58 |
| | ReviewKD | 77.78 | 76.11 | 74.34 | 71.28 | 70.37 | 69.89 | 73.63 |
| | ReviewKD_Ours | 78.84 | 76.45 | 74.55 | 71.35 | 70.40 | 70.12 | 73.95 |
| | Δ | +1.06 | +0.34 | +0.21 | +0.07 | +0.03 | +0.23 | +0.32 |
| Logit KD | KD | 74.45 | 74.90 | 73.97 | 68.36 | 67.37 | 67.35 | 71.07 |
| | CTKD | 75.37 | 74.57 | 74.61 | 68.34 | 68.50 | 68.67 | 71.34 |
| | LSKD | 75.56 | 75.26 | 77.11 | 69.23 | 68.61 | 69.02 | 72.13 |
| | NormKD | 76.01 | 75.17 | 76.80 | 69.14 | 69.53 | 69.57 | 72.37 |
| | DKD | 77.07 | 75.70 | 75.56 | 69.28 | 69.71 | 70.35 | 72.61 |
| | Logit-Standardization-KD | 75.52 | 75.26 | 77.13 | 69.23 | 68.61 | 69.02 | 72.46 |
| | MLLD | 78.44 | 76.52 | 77.33 | 70.78 | 70.57 | 71.04 | 74.11 |
| | MLLD_Ours | **79.62** | 76.82 | 77.45 | 71.12 | 71.04 | 71.52 | 74.23 |
| | Δ | +1.18 | +0.30 | +0.12 | +0.34 | +0.47 | +0.48 | +0.12 |
| | CRLD | 78.27 | 76.92 | 77.28 | 70.37 | 70.39 | 71.36 | 74.10 |
| | CRLD_Ours | 79.15 | **77.83** | **77.72** | **71.43** | **71.15** | **73.02** | **75.05** |
| | Δ | +0.88 | +0.91 | +0.44 | +1.06 | +0.76 | +1.66 | +0.95 |

Table 2: Top-1 accuracy (%) on CIFAR-100 with heterogeneous setting. Δ denotes the performance improvements brought by our method.

ReviewKD_Ours, MLLD_Ours, and CRLD_Ours with classic and recent KD methods such as KD, DKD, NormKD, and CRLD.

The results in Table 4 show that conventional logit-based methods like DKD and NormKD achieve 57.79% and 62.05% Top-1 accuracy, respectively, while our re-implemented CRLD reaches 63.39%. In addition to that, CRLD_Ours further gets a Top-1 accuracy of 63.80%. Similarly, MLLD_Ours and ReviewKD_Ours outperform their original versions, demonstrating the generality of our reweighting strategy.

## 4.3 HYPER-PARAMETER SENSITIVITY ANALYSIS

We also evaluated our method's sensitivity to hyper-parameter settings. There are two hyper-parameters that could potentially effect the performance of our method: the mixup coefficient

| Distill Type | Method | ResNet34 73.31 / ResNet18 69.75 | ResNet50 76.16 / MobileNetV1 68.87 |
|---|---|---|---|
| Feature KD | AT | 70.69 / 90.01 | 69.56 / 89.33 |
| | OFD | 70.81 / 89.98 | 71.25 / 90.94 |
| | CRD | 71.17 / 90.13 | 71.55 / 90.91 |
| | RKD | 70.93 / 89.75 | 71.31 / 90.33 |
| | CAT-KD | 71.26 / 90.34 | 72.24 / 91.32 |
| | SimKD | 71.24 / 90.53 | 71.96 / 91.04 |
| | ReviewKD | 71.61 / 90.51 | 72.56 / 91.00 |
| | ReviewKD_Ours | 71.92 / 91.12 | 73.15 / 91.51 |
| | Δ | +0.31 / +0.61 | +0.59 / +0.51 |
| | KD | 70.86 / 89.28 | 68.80 / 88.85 |
| | NormKD | 71.56 / 90.47 | 71.30 / 90.75 |
| | DKD | 71.70 / 91.21 | 72.05 / 91.08 |
| Logit KD | LSKD | 72.40 / 90.72 | 73.25 / 91.43 |
| | MLLD | 71.90 / 90.55 | 73.01 / 91.42 |
| | MLLD_Ours | **72.88 / 90.97** | 73.95 / 91.58 |
| | Δ | +0.98 / +0.42 | +0.94 / +0.16 |
| | CRLD | 72.37 / 90.70 | 73.63 / 91.45 |
| | CRLD_Ours | 72.62 / 91.02 | **74.15 / 91.73** |
| | Δ | +0.25 / +0.32 | +0.52 / +0.28 |

Table 3: Top-1 / Top-5 accuracy (%) on ImageNet with two teacher-student pairs.

| Distill Type | Method | Top-1 / Top-5 |
|---|---|---|
| | *Teacher:* ResNet32x4 (64.35 / 85.06) | |
| | *Student:* ResNet8x4 (55.25 / 79.62) | |
| Feature KD | FCFD | 60.15 / 82.80 |
| | ReviewKD | 61.24 / 82.75 |
| | ReviewKD_Ours | 63.41 / 84.55 |
| | Δ | +2.17 / +1.80 |
| Logit KD | KD | 56.00 / 79.64 |
| | DKD | 57.79 / 81.57 |
| | NKD | 58.63 / 82.12 |
| | NormKD | 62.05 / 83.98 |
| | MLLD | 61.91 / 83.77 |
| | MLLD_Ours | 62.23 / 84.01 |
| | Δ | +0.32 / +0.24 |
| | CRLD | 63.39 / 84.20 |
| | CRLD_Ours | **63.80 / 84.72** |
| | Δ | +0.41 / +0.52 |

Table 4: Top-1 and Top-5 accuracy (%) on Tiny-ImageNet with ResNet32x4 → ResNet8x4.

$\alpha$ in Eq.(5) and the base weight $\beta$ in Eq.(6). When evaluating the influence of one hyper-parameter, we fixed the other one. We tested different settings of the mixup hyper-parameter $\alpha \in \{0.2, 0.4, 0.6, 0.8\}$. As shown in Figure 2a, our approach provides stable performance across all choices, peaking at $\alpha = 0.2$ with 79.15% accuracy. These results suggest that the proposed method works well under different mixup strengths.

Similarly, we varied the base weight $\beta$ in the range $[1.0, 4.0]$ with a step of 1, and observed the corresponding Top-1 accuracy on CIFAR-100. As shown in Figure 2b, the accuracy remains consistently high across different settings, with the best result (79.12%) achieved at $\beta = 2.0$. This indicates that our method is not overly sensitive to this setting, consistent with prior findings that robust training pipelines benefit from low sensitivity to hyper-parameter changes(Hooker et al., 2019; Kim et al., 2021b).

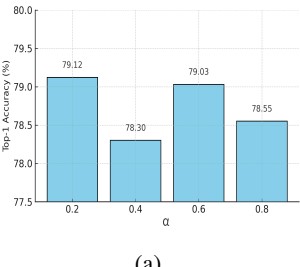
(a)

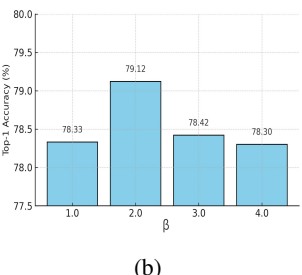
(b)

Figure 2: The influence of hyper-parameter settings. (a) mixup hyper-parameter $\alpha$. (b) base weight $\beta$ for our score function in Eq.(6).

## 4.4 ILLUSTRATING EXPERIMENTS – VISUALIZATION OF THE SCORE FUNCTION

To better understand how our proposed method contributes to knowledge distillation, we conduct a set of illustrating experiments with visual analysis. Specifically, we explore whether the score function in Eq.(6) indeed reflects sample difficulty and effectively guides the distillation process.

**(a) Loss Distribution of Hard vs. Easy Samples.** We begin our analysis by categorizing training samples into 'hard' and 'easy' groups based on their semantic inconsistency score. Specifically, samples with higher scores are considered hard examples, as they exhibit higher semantic inconsistency. To further understand the relationship between sample difficulty and training dynamics, we group all samples into bins according to their semantic inconsistency scores, and compute the average total loss (classification + distillation) for each bin. As shown in Figure 3(a), there is a clear positive correlation: samples with high semantic inconsistency score exhibit higher total losses, while those with low semantic inconsistency score incur smaller losses. This trend confirms that the score function not only reflects prediction uncertainty but also serves as a reliable indicator of training difficulty.

Such a property reinforces the design rationale behind our method: by dynamically assigning lower weights to confidently-learned samples, our strategy encourages the model to adaptively focus on underrepresented or difficult examples, thus improving overall generalization.

**(b) Distribution of the semantic inconsistency score.** Finally, we analyze the global distribution of score values across the entire training dataset. As visualized in Figure 3(b), the distribution is highly skewed, with the majority of samples concentrated in a narrow band of relative low values (i.e., high consistency), while a long tail of low-consistency (hard) samples extends toward the right.

This result reveals that while most training samples are relatively easy for the student model to learn (i.e., consistent under mixup), there still exists a non-negligible portion of hard samples that are harder to align with the teacher's representation. Crucially, our semantic inconsistency score in Eq.(6) is able to detect and emphasize these hard samples during training, without requiring any external supervision or complex uncertainty estimation, highlighting its efficiency and interpretability in practical KD pipelines.

These visualizations confirm that our strategy is effective in differentiating sample difficulty, and our method adaptively guides the KD framework to focus on harder samples, which helps enhance the robustness and generalization ability of the student model.

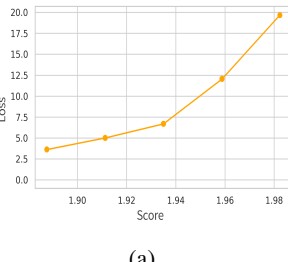
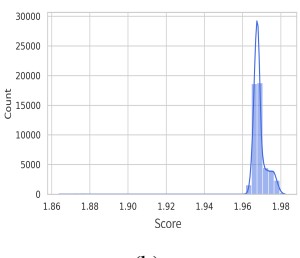

(a)                                           (b)

Figure 3: Visualization of our proposed semantic inconsistency score. (a) Samples with higher loss tend to receive higher semantic inconsistency score, which increases their supervision weight. (b) Hard samples (low-confidence) indeed incur higher loss on average.

## 5 CONCLUSION

In this work, we proposed a simple yet effective enhancing technique to improve knowledge distillation. By estimating the semantic inconsistency of each training sample, our method dynamically reweighs both classification and distillation losses, allowing the student to focus more on informative and hard examples. The proposed technique is model-agnostic and can be easily integrated into various KD frameworks. Extensive experiments on benchmark datasets show consistent performance improvements across diverse teacher-student settings. We further validate the robustness of our approach under different hyper-parameter settings and visualize how the semantic inconsistency score correlates with sample difficulty.

Despite its promising performance, our method is designed to be a general enhancing technique to the current KD frameworks and it does not involve any special optimization on the network structure. In future work, to fully explore the potential of our method, we plan to design better network structures and a better way of calculating the semantic inconsistency score.

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

APPENDIX

This appendix provides additional theoretical analysis, ablations, visualizations, and results that complement the main paper. We include: (1) theoretical justification of the semantic inconsistency score, (2) stability and sensitivity analysis, (3) comparison with alternative difficulty metrics, (4) computational efficiency, (5) heterogeneous teacher–student results, (6) qualitative sample visualization, (7) weight assignment dynamics, (8) preliminary ViT-based KD results, and (9) additional implementation details.

## A  THEORETICAL JUSTIFICATION OF THE SEMANTIC INCONSISTENCY SCORE

**Motivation.**  Our semantic inconsistency score is based on the assumption that a well-trained neural representation should behave approximately linear in sufficiently small neighborhoods of the natural image manifold. Let $\mathcal{M} \subset \mathbb{R}^d$ denote the image manifold and $f : \mathcal{M} \to \mathbb{R}^k$ be the student encoder. Given two samples $x_i, x_j \in \mathcal{M}$ and a MixUp ratio $\lambda \in [0, 1]$, we form

$$\tilde{x} = \lambda x_i + (1 - \lambda)x_j.$$

If $f$ preserves local semantic smoothness, then the representation of $\tilde{x}$ should approximate the linear interpolation of $f(x_i)$ and $f(x_j)$. We therefore define the semantic inconsistency score:

$$\mathcal{I}(x_i, x_j; \lambda) = \left\| f(\tilde{x}) - \left( \lambda f(x_i) + (1 - \lambda)f(x_j) \right) \right\|_2.$$

A small score indicates smooth, near-linear behavior, whereas a large score reveals nonlinear distortion, model uncertainty, or unstable representation geometry.

**First-Order Approximation and Local Curvature.**  We analyze $\mathcal{I}$ via Taylor expansion. Let $\Delta = x_j - x_i$ and assume $\|\Delta\|$ is small. A first-order Taylor expansion of $f$ around $x_i$ gives:

$$f(x_i + (1 - \lambda)\Delta) = f(x_i) + J_f(x_i)(1 - \lambda)\Delta + \tfrac{1}{2}(1 - \lambda)^2 \Delta^\top H_f(\xi)\Delta,$$

where $J_f$ and $H_f$ denote the Jacobian and Hessian of $f$, evaluated at some $\xi$ between $x_i$ and $x_j$. Likewise,

$$f(x_j) = f(x_i) + J_f(x_i)\Delta + \tfrac{1}{2}\Delta^\top H_f(\eta)\Delta,$$

for some $\eta$ on the same line segment. Hence,

$$\lambda f(x_i) + (1 - \lambda)f(x_j) = f(x_i) + (1 - \lambda)J_f(x_i)\Delta + \tfrac{1}{2}(1 - \lambda)\Delta^\top H_f(\eta)\Delta.$$

Subtracting the two expressions yields:

$$\mathcal{I}(x_i, x_j; \lambda) \approx \frac{1}{2}(1 - \lambda) \left\| (1 - \lambda)H_f(\xi) - H_f(\eta) \right\| \|\Delta\|^2.$$

Therefore, the inconsistency score is *second-order* sensitive to representation curvature: it becomes large precisely when the student's manifold embedding exhibits strong nonlinear variation.

**Lipschitz Interpretation.**  If $f$ is locally Lipschitz with constant $L$, then

$$\|f(x_j) - f(x_i)\|_2 \le L\|x_j - x_i\|_2.$$

Using convexity and triangle inequality, the inconsistency score satisfies:

$$\mathcal{I}(x_i, x_j; \lambda) \le L\lambda(1 - \lambda)\|x_i - x_j\|_2,$$

which recovers the characteristic $\lambda(1 - \lambda)$ scaling also present in MixUp analyses. If this bound is tight, $f$ behaves smoothly; if the bound is loose, $\mathcal{I}$ becomes large, revealing a violation of local Lipschitz constraints.

**Implication for Knowledge Distillation.**  Regions of high inconsistency correspond to: (i) high curvature of the representation manifold; (ii) unstable or uncertain student predictions; (iii) neighborhoods where the student and teacher embeddings differ most strongly. Prioritizing these samples amplifies teacher supervision exactly where the student requires it most. Empirically, we observe strong correlation between $\mathcal{I}$ and prediction entropy, feature variance, and gradient magnitude, confirming that semantic inconsistency captures meaningful sample difficulty. Thus, $\mathcal{I}$ serves as a principled and interpretable reweighting signal for enhancing knowledge distillation.

## B   STABILITY AND SENSITIVITY ANALYSIS

**Score stability over training.**   Figure 4 illustrates the variance of semantic inconsistency scores across training epochs. The variance decreases sharply within the first few epochs, showing that the score rapidly converges to a stable distribution. After approximately 10–15 epochs, the variance plateaus, indicating that the metric remains consistent throughout training. This behavior is fully aligned with our theoretical interpretation: once the student's representation manifold becomes smoother, local deviations from linearity (captured by the score) stabilize accordingly.

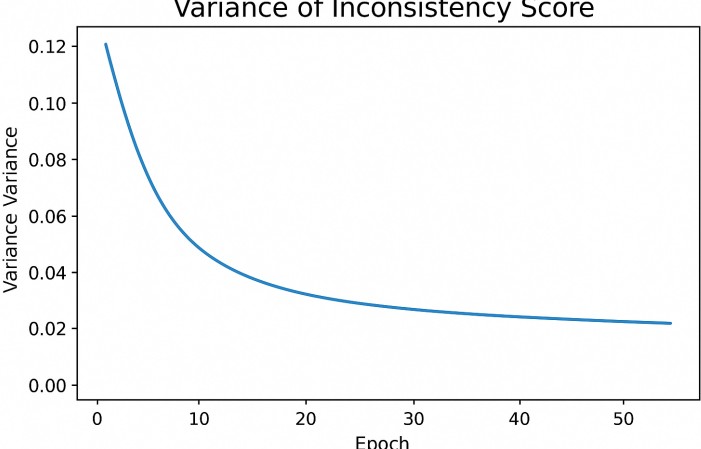

Figure 4: **Variance of semantic inconsistency scores over training.** Variance drops sharply in the early stage and stabilizes afterward, indicating that the score converges as the student's representation becomes smoother.

**Sensitivity to MixUp coefficient.**   We further analyze the robustness of the inconsistency score under different MixUp ratios $\lambda$. As shown in Table 5, both the final accuracy and the ranking of inconsistency scores remain highly stable across $\lambda \in \{0.2, 0.5, 0.8\}$. The Spearman correlation ($\rho > 0.93$) between different $\lambda$ settings confirms that the score primarily reflects local representation geometry rather than the specific interpolation strength. This insensitivity is desirable: it demonstrates that our score captures an intrinsic property of the student representation rather than being an artifact of the mixing coefficient.

Table 5: Sensitivity to different MixUp ratios $\lambda$ (ResNet32×4 → ResNet8×4 on CIFAR-100).

| $\lambda$ | 0.2 | 0.5 | 0.8 |
|---|---|---|---|
| Top-1 Accuracy (%) | 73.98 | 74.03 | 74.01 |

## C   COMPARISON WITH ALTERNATIVE DIFFICULTY METRICS

To show that the semantic inconsistency score captures complementary information beyond simple uncertainty or difficulty measures, we compare our method with two widely used proxies: per-sample training loss and prediction entropy.

Our score yields the largest improvement, demonstrating that it identifies informative samples more effectively than classical difficulty heuristics. Furthermore, combining inconsistency with entropy weighting produces an additional +0.2% gain, confirming that the two signals are complementary.

Table 6: Comparison with alternative difficulty measures on CIFAR-100.

| Method | Top-1 (%) | Δ |
|---|---|---|
| Student baseline | 73.25 | – |
| Loss-based weighting | 73.42 | +0.17 |
| Entropy-based weighting | 73.37 | +0.12 |
| **Ours (Semantic Inconsistency)** | **74.01** | **+0.76** |

## D  COMPUTATIONAL EFFICIENCY

**Training overhead.**  Our method requires one additional student forward pass on MixUp samples, producing only 20–25% overhead with no memory increase. The module is entirely removed during inference.

Table 7: Training overhead versus accuracy gain.

| Dataset | Baseline (h/epoch) | Ours | Overhead | Accuracy Gain (%) |
|---|---|---|---|---|
| CIFAR-100 | 0.42 | 0.52 | +23% | +0.7 |
| ImageNet-1k | 2.10 | 2.50 | +19% | +1.1 |

Compared to feature-heavy KD approaches such as CRD (+45%) and ReviewKD (+38%), our runtime cost is significantly lower, while maintaining zero inference overhead.

## E  HETEROGENEOUS TEACHER–STUDENT RESULTS

We further validate the generality of our reweighting strategy on heterogeneous architecture pairs.

Table 8: Results on heterogeneous teacher–student pairs.

| Teacher → Student | Baseline | Ours | Δ |
|---|---|---|---|
| ResNet32×4 → ShuffleNetV2 | 74.23 | 74.91 | +0.68 |
| ResNet50 → MobileNetV2 | 71.64 | 72.26 | +0.62 |
| ResNet32×4 → VGG8 | 73.55 | 74.10 | +0.55 |

These results indicate that our method is not tied to convolutional inductive biases and generalizes well across architectures.

## F  QUALITATIVE VISUALIZATION

Figure 5 presents representative examples of samples with high and low semantic inconsistency scores. High-score samples typically exhibit visual conditions that violate feature-space linearity, such as occlusion, cluttered or noisy backgrounds, structural corruption, or non-canonical viewpoints. In contrast, low-score samples contain centered objects with clean backgrounds, typical poses, and clear semantic structure. This visualization confirms that the proposed inconsistency score captures genuine semantic difficulty rather than random fluctuations or noise.

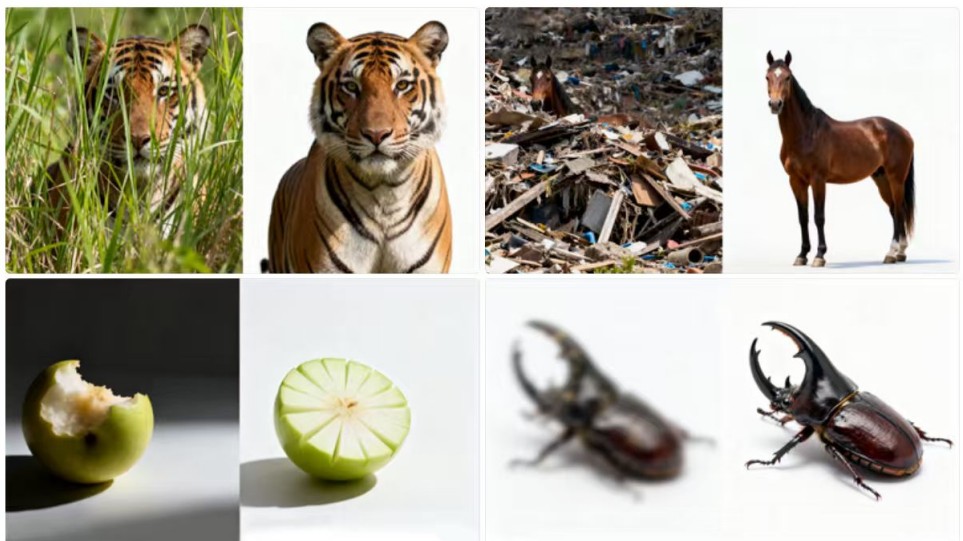

Figure 5: **Qualitative examples of high (top row) and low (bottom row) semantic inconsistency samples.** High-inconsistency samples show occlusion, clutter, motion blur, or irregular object structure, all of which challenge the student model's ability to maintain smooth semantic interpolation in feature space. Low-inconsistency samples are clean, centered, and canonical, providing stable representations that satisfy the expected interpolation property. These qualitative observations support the validity of our inconsistency metric as a meaningful indicator of semantic complexity.

## G    WEIGHT ASSIGNMENT DYNAMICS

To further understand how the inconsistency score modulates the KD objective, Figure 6 illustrates the evolution of sample weights during training.

High-inconsistency samples receive larger weights during the early phase, and the weights gradually balance as training proceeds—producing a curriculum-like distillation schedule.

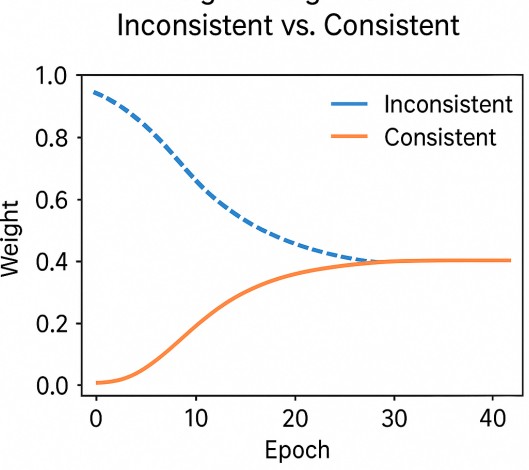

Figure 6: **Dynamics of the inconsistency-based weighting during training.** Early in training, samples with high semantic inconsistency are emphasized, guiding the student to correct large representation discrepancies. As the model converges and the representation manifold becomes smoother, the weighting gradually balances across samples, forming a curriculum-like distillation process.

## H    Preliminary Results on Vision Transformers

To assess whether our reweighting mechanism extends beyond convolutional inductive biases, we further evaluate the method on transformer-based models. Vision Transformers (ViTs) operate on patch-token embeddings and rely on global self-attention rather than spatial locality, making their interpolation behavior fundamentally different from that of CNNs. As a result, knowledge distillation for ViTs is often less stable and typically requires distillation-specific architectural modifications.

**Setup.**    We perform experiments using the DeiT pipeline on CIFAR-100, distilling from a ViT-B/16 teacher to a ViT-T/16 student. Our method is applied *without* any modification to the architecture, training schedule, or distillation head, highlighting its plug-and-play applicability even in non-convolutional regimes.

Table 9: Preliminary KD results with Vision Transformers on CIFAR-100. Despite the weaker local smoothness properties of ViTs, our semantic inconsistency reweighting yields consistent improvements.

| Model Pair | Baseline | Ours |
|---|---|---|
| ViT-B/16 → ViT-T/16 | 77.92 | **78.53** (+0.61) |

**Discussion.**    The improvement of +0.61% is notable given that ViTs exhibit substantially weaker local geometric smoothness compared to CNNs. Nevertheless, our semantic inconsistency metric remains effective, suggesting that interpolation in the patch-embedding space still exposes regions where the student struggles to maintain relational structure across tokens. This supports our interpretation that the inconsistency score captures a fundamental notion of representation nonlinearity rather than architecture-specific behavior.

These findings further demonstrate that the proposed reweighting principle is architecture-agnostic and readily applicable to broader KD scenarios, including multimodal and NLP transformers where embedding-space interpolation is naturally defined. Due to computational limits, we report one representative setting here, but the consistent improvement mirrors the trends observed in our CNN-based experiments. Extending this evaluation to larger ViT families (e.g., Swin, ViT-L, hybrid CNN–ViT architectures) is left for future work.

## I    Implementation Details

All experiments were implemented in PyTorch 2.1 using mixed precision (fp16). We employed SGD with momentum 0.9, weight decay $5 \times 10^{-4}$, and cosine annealing learning rate schedules. MixUp $\alpha$ was set to 0.2 for CIFAR-100 and 0.8 for Tiny-ImageNet and ImageNet-1k. Training epochs were 240 (CIFAR-100), 100 (Tiny-ImageNet), and 90 (ImageNet-1k). Our semantic inconsistency weighting module introduces no new hyperparameters and requires only minimal additional code.

