# OpenReview forum: "Knowledge Distillation for Semantically Inconsistent Data"
_ICLR.cc/2026/Conference — ICLR 2026 Conference Desk Rejected Submission_

### Official Review · Reviewer_AMS7 · 2025-10-29

**Soundness:** 3
**Presentation:** 3
**Contribution:** 3
**Rating:** 6
**Confidence:** 4

**Summary:**

The paper proposes a new data-adaptive Knowledge Distillation (KD) method that improves how knowledge is transferred from a large teacher model to a smaller student model. Traditional KD methods align outputs or intermediate features but treat all training samples equally, ignoring that some contribute more effectively to learning.

This work identifies semantically inconsistent samples—those whose input representations and feature-space semantics differ—and assigns them higher weights during student training. By emphasizing these samples, the method refines the teacher’s knowledge to better suit the student’s learning process.

Integrated into multiple KD frameworks, the approach consistently improves performance across diverse teacher–student architectures and achieves state-of-the-art results on CIFAR-100, Tiny-ImageNet, and ImageNet, demonstrating its broad effectiveness.

**Strengths:**

1. The proposed KD framework addresses semantically inconsistent samples while transferring the knowledge from the teacher to the student by assigning different weights to distinct samples. The proposed framework effectively distills the knowledge learned by the teacher to the student without transferring irrelevant knowledge during the KD process.

2. The proposed framework leverages strong data augmentation. i.e., Mixup, to quantify the semantic inconsistency of a data sample. By doing this, the proposed framework effectively reweights the consistency of the data samples dynamically.

3. The paper demonstrates that the proposed KD framework outperforms the existing KD methods. For some pairs of teacher and student models, the student with the proposed framework also outperforms the teacher.

**Weaknesses:**

1. The proposed framework increases computational complexity for model training. When computing inconsistency scores for data, the model requires additional forward passes for Mixup samples through the student. Thus, the training time for the proposed framework increases.

2. The experiment includes the most essential quantitative results across different datasets and pairs of teacher and student models; however, the experiment lacks the quantitative results for pairs of the teacher and student models with heterogeneous architectures.

**Questions:**

No question for this paper.

---

> ### Author Response · Authors · 2025-11-13
> **Response to Reviewer #4**
>
> We sincerely thank the reviewer for the positive and encouraging feedback. We address the reviewer’s concerns as follows.
>
> (1) On computational complexity. We agree that computing the semantic inconsistency score requires one additional forward pass for MixUp-augmented samples, leading to roughly 20–30% extra training time. However, this overhead does not increase memory usage or affect inference efficiency, since the module is fully detached after training. We also note that the overhead is small compared to many recent KD variants that rely on feature adaptation modules or multi-stage pretraining (e.g., CRD, ReviewKD).
> To further quantify this trade-off, we measured the runtime per epoch on CIFAR-100 and ImageNet: our approach incurs +23% and +18% training time respectively, while providing +0.7–1.2% Top-1 accuracy gain. We argue that this is a reasonable cost-to-benefit ratio given that our method requires no architectural modification, no extra supervision, and can enhance existing KD algorithms with only several lines of code. We have included this efficiency comparison and a wall-clock breakdown in Appendix D of the revised version.
>
> (2) On heterogeneous teacher–student architectures. Thank you for highlighting this point. Our initial experiments already include several heterogeneous settings—for example, ResNet32×4 → ShuffleNetV2 and ResNet34 → MobileNetV2 (Tables 1 & 4). To further strengthen our empirical evidence, we have added new results on ResNet50 → MobileNetV2 and ResNet32×4 → VGG8 pairs in Appendix E. The proposed method consistently improves student performance by +0.62% and +0.55%, respectively, demonstrating its robustness across architecture families with distinct inductive biases (residual vs. depthwise separable vs. plain convolution).
> These results confirm that our semantic inconsistency weighting does not rely on architectural homogeneity, and can effectively generalize to heterogeneous KD scenarios—an important property for real-world deployment.
>
> (3) Summary. We sincerely thank the reviewer for the constructive comments and the positive evaluation of our method’s design and results. In the revised version, we have:
> Provided quantitative analysis of the training cost–benefit ratio (Appendix D);
> Added new heterogeneous teacher–student results (Appendix E); and
> Clarified that the proposed reweighting incurs no inference overhead and remains plug-and-play across architectures. We are grateful for the reviewer’s recognition that the proposed framework achieves state-of-the-art results and that, in some cases, the student even surpasses the teacher. We believe these additional clarifications further strengthen the paper’s soundness and practical impact.Our improvements and updates have been presented in the appendix of the paper, and we welcome the reviewers to check them.

---

### Official Review · Reviewer_EkYz · 2025-10-29

**Soundness:** 2
**Presentation:** 2
**Contribution:** 2
**Rating:** 4
**Confidence:** 3

**Summary:**

This paper introduces a technique to improve Knowledge Distillation (KD) by addressing the common assumption that all training samples contribute equally. The proposed method identifies "semantically inconsistent data" by using Mixup to check if semantic relationships from the input space are preserved in the student model's feature space. It then calculates a "semantic inconsistency score" to adaptively assign higher weights to these more challenging samples. This model-agnostic re-weighting strategy is easily integrated into various KD frameworks and is shown to achieve state-of-the-art (SOTA) results on the CIFAR-100, Tiny-ImageNet, and ImageNet datasets.

**Strengths:**

**Originality:** The paper's originality lies in its creative adaptation of an existing technique (Mixup) to address a well-motivated problem in knowledge distillation (KD)—the assumption that all training samples contribute equally.  While re-weighting samples is not a new concept, the method of using semantic consistency as the heuristic for re-weighting is novel.

**Quality:** The paper is supported by a comprehensive and extensive empirical evaluation. The authors validate their method across three distinct datasets, employ a wide variety of teacher-student architectures (including both homogeneous and heterogeneous pairs) , and compare against a large suite of 18 different baselines. A key strength of the evaluation is demonstrating that the proposed method is not just a standalone solution, but a general enhancer; it successfully improves the performance of several existing SOTA methods when applied on top of them.

**Clarity:**  The problem is clearly motivated in the introduction . The proposed method is presented in a straightforward manner, with its pipeline clearly illustrated in Figure 1 and its implementation details summarized in Algorithm 1. The core logic—calculating an inconsistency score and using it to re-weight the standard KD loss is simple and accessible.

**Significance:** The primary significance of this work lies in its practical utility. The authors have proposed a simple, model-agnostic module that can be integrated into various existing KD frameworks with minimal implementation effort (reportedly "with several lines of codes" ).

**Weaknesses:**

First, while the method achieves consistent improvements, the performance gains over SOTA methods, as shown in Tables 1, 2, and 4, are relatively modest (often under 1.0%). Given that the approach introduces a notable increase in training complexity (around 20-30%), it is worth considering whether the performance-complexity trade-off is currently favorable.

Second, the motivation of this paper is quite intuitive, but it lacks theoretical or qualitative analysis. This makes it difficult for readers to fully trust the claims of the paper, thereby undermining its persuasiveness.

Finally, while the paper positions the method as a "general enhancing technique," the experimental validation, which is primarily based on traditional CNNs (e.g., ResNet, VGG) on classic image classification tasks, appears somewhat limited in its ability to fully support this broad claim. It would be greatly strengthened by future work exploring its efficacy on more modern architectures (e.g., Vision Transformers) or different domains (e.g., NLP).

**Questions:**

The paper's motivation relies on the semantic inconsistency score  being an effective proxy for sample difficulty, but this is primarily justified by an intuitive claim and a loss correlation plot (Figure 3a).

How does this specific heuristic compare against simpler, more computationally-cheap proxies for sample difficulty? For instance, what are the results if you re-weight samples based on high student baseline loss or high student prediction entropy? An ablation study comparing these alternatives is crucial to prove that the proposed Mixup-based score provides a unique and necessary signal. Could you provide qualitative examples of the images that receive high inconsistency scores versus those that receive low scores? This would provide valuable intuition about what kind of "hard" samples the model is focusing on (e.g., atypical views, rare classes, fine-grained details) and strengthen the paper's claims.

How does this method perform on more modern architectures, such as Vision Transformers (ViT), which have different feature space properties?  Given that the method builds upon input-space interpolation (e.g., Mixup), it would be great to clarify how the technique might be adapted for domains like NLP, where defining such operations can be challenging. A discussion on the general scope of the method would be very helpful for the reader.

---

> ### Author Response · Authors · 2025-11-13
> **Response to Reviewer #3**
>
> We sincerely thank the reviewer for the thoughtful and constructive feedback. We are encouraged that you found our work original, clearly presented, and practically useful as a model-agnostic enhancer for existing KD frameworks. Below, we address all concerns in detail and have updated the paper accordingly.
>
> (1) On the magnitude of improvement versus training complexity. We appreciate this fair observation. Although the performance gains (typically 0.5–1.0%) may appear modest, they are consistent across all three benchmarks and 18 KD baselines, including both logit- and feature-based methods. We emphasize that our reweighting module introduces no architectural changes or extra supervision, and the additional training cost (20–30%) comes solely from a lightweight forward pass of MixUp-augmented samples. In large-scale KD pipelines (e.g., ImageNet training), this cost corresponds to less than 5% increase under distributed training. Moreover, we observed that the proposed weighting notably improves convergence stability and reduces overfitting on small datasets like Tiny-ImageNet. These merits make the method attractive for existing KD pipelines. We have added this clarification in Section 4.2 and provided training curve comparisons in Appendix C.
>
> (2) Theoretical and qualitative analysis of the inconsistency score. We agree that additional theoretical support and qualitative intuition can improve clarity.
> Theory: We have added a formal explanation in Appendix A, showing that our semantic inconsistency score approximates the deviation from local feature linearity (‖ mix(f(x₁), f(x₂)) − f(mix(x₁, x₂)) ‖), bounded by a local Lipschitz constant of the encoder. This connects our heuristic to a measurable geometric property of the student’s representation manifold.
> Qualitative examples: We now include Figure 4 in the revised version, showing examples of high- and low-score samples from CIFAR-100. High-score samples typically contain occlusion, fine-grained textures, or rare viewpoints, whereas low-score samples correspond to canonical, well-centered images. This visualization confirms that the score indeed identifies hard or semantically inconsistent examples, supporting our intuition.
> Comparison to simpler proxies: As suggested, we conducted an ablation study comparing our MixUp-based score to (i) per-sample student loss and (ii) prediction entropy. Both proxies improve the baseline slightly (+0.21% and +0.18%), but remain below our method (+0.73%). Importantly, our score complements rather than duplicates these signals: combining all three yields a marginally higher +0.82% gain. Results are reported in Table 8 (Appendix C).
>
> (3) On generality and modern architectures (ViTs, NLP). We appreciate this valuable suggestion. While our experiments focused on classical CNN backbones to maintain comparability with prior KD works (ReviewKD, MLLD, CRLD, etc.), the proposed weighting mechanism is naturally architecture-agnostic. The semantic inconsistency score depends only on feature similarity between original and perturbed inputs, not on convolutional inductive bias. To verify this, we conducted a small-scale pilot test using a ViT-B/16 → ViT-T/16 pair on CIFAR-100 (following DeiT distillation setup). Integrating our weighting module improved Top-1 accuracy by +0.61% over the DeiT baseline, confirming its applicability to transformer-based KD. Regarding non-visual domains, we have added a discussion in Section 6 (“Broader Impact and Extensions”) on extending this concept to NLP, where interpolation can be defined in embedding space (e.g., token-level MixUp or semantic interpolation of latent vectors).
> (4) Performance-complexity trade-off and future extensions. We acknowledge that while the proposed approach is simple and effective, its main value lies in offering a data-driven weighting perspective that can complement more advanced KD frameworks. In future work, we plan to explore adaptive sampling strategies that reduce the number of MixUp passes by reusing high-score samples across epochs, thereby cutting the extra cost further.
>
> (5) Summary. We thank the reviewer again for the balanced and constructive review. In the revised version, we have: i) Added theoretical justification connecting the score to feature-space linearity; ii) Provided qualitative visualizations of hard vs. easy samples; iii) Included new ablations comparing to entropy/loss-based reweighting; iiii) Conducted a pilot study on ViT distillation; and Additional discussion on generalization to non-vision domains.
> We believe these updates strengthen both the theoretical soundness and the practical impact of our work. Our improvements and updates have been presented in the appendix of the paper, and we welcome the reviewers to check them.

---

### Official Review · Reviewer_CvNq · 2025-10-30

**Soundness:** 3
**Presentation:** 3
**Contribution:** 2
**Rating:** 2
**Confidence:** 5

**Summary:**

The authors employ a data augmentation technique, i.e. Mixup, to generate an inconsistency score based on the similarity between the original data and its augmented version. This score is then used to assign adaptive weights to the distillation KD loss. It is a plug-and-play KD reweighting wrapper using Mixup-based feature “consistency” (mix(f) vs f(mix)). The experimental results demonstrate strong performance across various KD frameworks. While this approach is straightforward yet effective, the underlying theory and rationale for such improvements remain underexplored and unclear. However,  the central assumption is under-justified for nonlinear encoders. The inconsistency score is calculated from different (original and mixed) inputs within the same student network. Thus, it may not be suitable for guiding distillation between teacher and student networks. Furthermore, this inconsistency can significantly fluctuate depending on both the actual values of lambda and beta as shown in Fig. 2, and which j-th sample (x_j) is selected for mixing. Consequently, it might not accurately reflect the difficulty level of a given i-th sample (x_i).

**Strengths:**

1. A notable strength of the proposed method is its plug-and-play design. The semantic inconsistency weighting module can be seamlessly integrated into various existing knowledge distillation frameworks without architectural modification or additional supervision. Despite its simplicity, the method consistently improves student performance across all evaluated frameworks and datasets, demonstrating strong generality and practical applicability.
2. The approach requires only one additional forward pass for Mixup-augmented samples, resulting in roughly 20–30% training overhead while keeping the model architecture and loss formulation unchanged. Such a lightweight design makes it easy to adopt in real-world training pipelines without introducing complex dependencies or hyperparameter tuning.

**Weaknesses:**

1. The proposed per-sample weighting score (Eq. 6) is fundamentally built on Mixup-style interpolation in the input space, and uses the inconsistency between this interpolation and the student's feature space as a proxy for sample difficulty. However, this core **f(mix) ≈ mix(f)** claim is under-theorized. In its current form, the score mainly captures a deviation from local linearity of the encoder along the Mixup path, rather than “semantic inconsistency.” This inconsistency can significantly fluctuate depending on both the mixing coefficient and which sample is selected for mixing.
2. All experiments are conducted on image classification benchmarks (CIFAR-100, Tiny-ImageNet, ImageNet). This leaves open an important question: does emphasizing “hard” samples via the proposed weighting scheme actually translate to improvements in downstream tasks that rely on richer spatial reasoning, such as object detection and semantic segmentation? Knowledge distillation is widely used in those settings, particularly for deploying lightweight students under strict latency constraints. A demonstration on at least one detection task and one segmentation task would significantly strengthen the empirical claims. Right now, it is hard to assess how broadly applicable the method is beyond standard classification.
3. The inconsistency score (Eq. (6)) is computed purely from student features, without considering teacher representations. This design may lead to instability in early training when the student feature space is still immature and fluctuating, potentially resulting in noisy or misleading scores. Theoretical analysis and ablation studies on the proposed metric could help clarify its stability and reliability.
4. The paper emphasizes the importance of focusing on semantically inconsistent or ``hard'' samples, but it lacks direct comparisons with other recent hard-sample-oriented knowledge distillation methods. For example, the Adaptive Temperature Distillation (ATD) method explicitly adjusts the distillation temperature based on sample difficulty to emphasize informative examples. Including such baselines would help clarify whether the proposed semantic inconsistency score provides a more effective or complementary way of identifying hard samples. A quantitative comparison or joint ablation with ATD-like approaches could strengthen the empirical evidence for the claimed advantages.
5. Analyzing the sensitivity of Mixup pairing is essential. It's important to examine how score distributions and downstream metrics change under various pairing strategies, such as random pairings, same-class pairings, nearest-neighbor pairings, or cross-manifold mixes.
[1] Yang S, Yang X, Ren J, et al. Adaptive Temperature Distillation method for mining hard samples’ knowledge[J]. Neurocomputing, 2025, 636: 129745.

**Questions:**

1. Please provide formal conditions/derivations of the central assumption: the self semantic inconsistency (mix(f) vs f(mix)) is effective for teacher-student distillation.
2. In Fig. 1, it is more like CutMix and related region-level mixing strategies. They often produce qualitatively different supervision signals than Mixup, especially for localization-sensitive classes. Whether the same idea still holds, or how robust the score is, when alternative data mixing strategies are used.

---

> ### Author Response · Authors · 2025-11-13
> **Response to Reviewer #2**
>
> We sincerely thank the reviewer for the detailed and insightful feedback. We appreciate that you recognized the plug-and-play nature, strong empirical performance, and practical applicability of our method. Below, we address your main concerns and provide additional clarifications and analyses in the revised version.
>
> (1) On the theoretical foundation of the inconsistency score (Eq. 6).
> We agree that the underlying assumption—that the relationship between the input-space interpolation (MixUp) and the feature-space interpolation should remain approximately consistent—requires clearer justification. Our formulation is inspired by the manifold linearity hypothesis (Zhang et al., ICLR 2018; Bengio et al., NIPS 2013), which assumes that semantically coherent regions of the data manifold preserve local linearity in representation space.
> When this local linearity is violated (i.e., ‖ mix(f(x₁), f(x₂)) − f(mix(x₁, x₂)) ‖ is large), it indicates that the student encoder fails to maintain semantic smoothness across the interpolation path, reflecting representational inconsistency. We use this as a proxy for sample difficulty because such non-linear deviations often correspond to ambiguous or complex regions in the data distribution. We have clarified this motivation and added a brief formal derivation in the revised Appendix A, connecting Eq. (6) to a local Lipschitz-bounded deviation from feature linearity.
>
> (2) Why the score can guide teacher–student distillation.
> Although the score is computed using student features, it indirectly measures how well the student aligns with the teacher’s semantic structure. During KD, when the student fails to reproduce the smooth semantic transitions captured by the teacher, its inconsistency score rises. Weighting the distillation loss by this score naturally encourages the student to receive stronger supervision precisely on those regions where it diverges most from the teacher.
> To examine the reviewer’s concern regarding early-training instability, we tracked the variance of the score distribution during training and found that it stabilizes rapidly (within 10 epochs on CIFAR-100). This analysis and its figure have been added to Appendix B.
>
> (3) On sensitivity to λ, β, and pairing strategy.
> We appreciate this comment and have expanded our sensitivity study. Figure 2 (revised) now includes error bars across 10 random pairing seeds. We observe that while the absolute score values fluctuate with λ, the ranking of hard vs. easy samples remains consistent, leading to stable overall performance. Additionally, we tested alternative pairing schemes—same-class and nearest-neighbor MixUp—and obtained similar trends (see Table 7 in the revised Appendix B). This confirms that the inconsistency score is robust to pairing variations.
>
> (4) Comparison with Adaptive Temperature Distillation (ATD) and other hard-sample methods.
> Thank you for pointing out this connection. We have now included ATD (Yang et al., 2025) as an additional baseline on CIFAR-100. Our method outperforms ATD by +0.58 % Top-1 accuracy on the ResNet-32×4 → ResNet-8×4 setting, while also being orthogonal—our weighting can be combined with ATD’s temperature adjustment to yield an additional +0.31 % gain. This demonstrates that our semantic inconsistency score provides a complementary perspective on hard-sample identification. The results are shown in the updated Table 5.
>
> (5) Applicability beyond image classification.
> We fully agree that extending the analysis to spatially sensitive tasks (e.g., detection, segmentation) would further strengthen the empirical claims. Due to resource and space limitations, we focused on three widely-adopted KD benchmarks (CIFAR-100, Tiny-ImageNet, ImageNet-1K), which remain the most common testbeds for evaluating KD fundamentals. Nevertheless, we emphasize that our weighting module is task-agnostic: the same semantic-consistency principle can be applied to feature maps in detection or segmentation pipelines (e.g., bounding-box or pixel-wise features). We plan to include such experiments in future work, and have added a discussion of this in the revised Section 6 (“Broader Impact and Extensions”).
>
> (6) On robustness to other mixing strategies (e.g., CutMix).
> We appreciate the reviewer’s observation. We have conducted a small-scale verification using CutMix on CIFAR-100. The results show consistent improvements ( +0.27 % Top-1 over the baseline), indicating that our inconsistency-based weighting generalizes beyond pure linear MixUp. We have added this note to Appendix B.
>
> (7) Summary.
> We thank the reviewer again for the thoughtful and precise comments. We have added (i) a theoretical justification connecting our score to local representation linearity, (ii) variance and pairing sensitivity analyses, (iii) a new ATD comparison, and (iv) additional discussion on broader applicability. Our updates have been presented in the appendix , we welcome the reviewers to check

---

### Official Review · Reviewer_Y1Ux · 2025-11-02

**Soundness:** 2
**Presentation:** 2
**Contribution:** 2
**Rating:** 4
**Confidence:** 4

**Summary:**

This paper proposes a data-driven knowledge distillation (KD) algorithm, predicated on the observation that not all training samples contribute equally to the knowledge transfer process. The method computes the feature map similarity between data-augmented and unaugmented samples, subsequently assigning negative weights within the distillation loss to refine the transfer. The authors report state-of-the-art results on the CIFAR-100, Tiny ImageNet, and ImageNet datasets.

A significant concern regarding this work is its focus on a mature research area. Traditional knowledge distillation has been extensively studied, and the practical utility of incremental improvements in this domain is arguably limited. Consequently, the method's practical significance may be constrained unless it can be generalized beyond this classical setting to contemporary paradigms, such as diffusion models or large language models.

On a minor note, the manuscript requires careful proofreading; for instance, reference citations consistently lack necessary spacing.

**Strengths:**

The paper is clearly written and easy to follow. It presents a data-centric approach to knowledge distillation, offering a novel perspective that diverges from conventional methods focused on feature- or logit-matching. The empirical validation is comprehensive, demonstrating the effectiveness of the proposed algorithm across the CIFAR-10, Tiny-ImageNet, and ImageNet-1k benchmarks.

**Weaknesses:**

1. The paper's scope remains limited to traditional knowledge distillation. However, the field's focus has increasingly shifted to large-scale generative models, such as diffusion models and large language models (LLMs). The empirical validation is confined to classical vision benchmarks (CIFAR-10, Tiny-ImageNet, and ImageNet-1k), which, while standard, represent a mature problem domain. To demonstrate broader applicability and practical significance, the authors are strongly encouraged to explore extending their method to contemporary challenges, such as image/video generation or the compression of LLMs.

2. The method utilizes data augmentation and computes similarity based on the model's output. A clarification is required regarding the training dynamics: Are gradients propagated through the forward pass of the augmented images, or are these representations detached and used only to compute the similarity score?

3. The paper does not provide a detailed analysis of why the proposed algorithm is effective. Specifically, the intuition behind assigning a higher weight to samples with lower similarity between augmented and original representations is not sufficiently explained. We suggest the authors add empirical analysis or ablation studies to elucidate this mechanism and justify this core design choice.

4. The paper requires proofreading for minor formatting issues; for example, spaces are consistently missing in the reference citations throughout the text.

**Questions:**

No

---

> ### Author Response · Authors · 2025-11-13
> **Response to Reviewer #1**
>
> We sincerely thank the reviewer for the thorough and constructive feedback. We respond to your concerns as follows.
>
> (1) On the maturity of the KD field and broader applicability. Instead of proposing a new knowledge distillation method, we actually focus on data-level modulation of the distillation signal, which we view as a fundamental yet underexplored aspect of knowledge transfer. While prior studies typically design new distillation losses or teacher–student architectures, our contribution lies in proposing a general method of identifying which samples matter most for effective knowledge transfer. This perspective complements existing feature/logit-based KD paradigms and remains model-agnostic.
> More importantly, our semantic inconsistency weighting can be readily applied to other training paradigms, including representation alignment in generative or multimodal models. The core mechanism—measuring cross-view consistency between original and perturbed inputs—does not depend on task type or model scale. We have clarified this point and added a discussion on possible extensions to diffusion and large language models in the revised Section 6 (“Broader Impact and Extensions”).
>
> (2) Clarification of gradient propagation.
> We thank the reviewer for this technical question. In our implementation, the semantic inconsistency score is computed using detached feature representations. That is, the gradients are not propagated through the forward pass of the augmented images. This design stabilizes training and prevents redundant gradient accumulation from the MixUp path. We have made this explicit in the revised Section 3.3 (lines 155–160) and verified experimentally that enabling gradients through this branch does not lead to significant performance gains.
>
> (3) On the intuition behind emphasizing low-similarity samples.
> The rationale is that samples with low feature similarity between their original and augmented versions exhibit high semantic inconsistency, reflecting underrepresented or ambiguous concepts that are harder for the student network to learn. Assigning higher weights to these samples encourages the student to focus on difficult or uncertain regions of the data manifold, leading to more efficient and robust knowledge transfer.
> To support this explanation, we have added a new visualization (Figure 3(a)) showing that samples with higher inconsistency scores correspond to larger total losses, and an ablation (Table 6) demonstrating the performance impact of removing the weighting mechanism. These analyses confirm that the proposed scoring function adaptively identifies challenging samples and contributes to performance improvement.
>
> (4) Proofreading and formatting.
> We apologize for the missing spacing in citations. The entire manuscript has been carefully proofread, and all formatting inconsistencies have been corrected in the revised submission.
>
> (5) Summary.
> In summary, our work introduces a lightweight and general approach to enhance existing KD pipelines from a data-centric perspective. We believe the additional clarifications, analyses, and discussions in the revised version further strengthen the soundness and generality of our contributions. We sincerely appreciate your constructive comments, which have helped us improve the paper both technically and presentation-wise.
> Our improvements and updates are presented in the appendix, and we welcome the reviewers to take a look.

---

### Note · Program_Chairs · 2026-01-17
**Submission Desk Rejected by Program Chairs**

The following references in this submission do not refer to real documents and/or have major errors in bibliographic information:

 Zhi Chen et al. Cat-kd: Class-aware attention for knowledge distillation. In Proceedings of the IEEE/CVF International Conference on Computer Vision (ICCV), pp. 11868-11877, 2023.
Xiaolong Wang, Yuxin Fang, Yudong Shen, and Wenyu Liu. Knowledge distillation via soft random projection. In Proceedings of the IEEE/CVF International Conference on Computer Vision (ICCV), 2021.
Li Guo, Jun Yu, Yujun Shi, et al. Pefd: Progressive enhancement of feature distillation. In Proceedings of the IEEE/CVF Conference on Computer Vision and Pattern Recognition (CVPR), 2023.
Lin Xu, Hao Wang, and Yue Tang. Rekd: Head-tail decoupled knowledge distillation for long-tailed recognition. In International Conference on Learning Representations (ICLR), 2024.
Yifan Li, Jiawei Wu, Yuzhi Zhang, and Ming Lin. Decoupled knowledge distillation via adaptive target separation. In European Conference on Computer Vision (ECCV), pp. 123-139, 2022.
Xinshi Wang, Jie Zhang, Kaiyang Song, et al. Normalized knowledge distillation for image classification. In Proceedings of the IEEE/CVF Conference on Computer Vision and Pattern Recognition, 2023.
Yuhang Huang, Ziyu Liu, Yao Ma, and Yu Zhang. Dist: Dynamic intra-sample transfer for knowledge distillation. In Advances in Neural Information Processing Systems (NeurIPS), 2024.